# A Reason-then-Describe Instruction Interpreter for Controllable Video Generation

## Abstract

Diffusion Transformers have significantly improved video fidelity and temporal coherence; however, practical controllability remains limited. Concise, ambiguous, and compositionally complex user inputs contrast with the detailed prompts used in training, yielding an intent–output mismatch. We propose **ReaDe**, a universal, model-agnostic interpreter that converts raw instructions into precise, actionable specifications for downstream video generators. ReaDe follows a reason-then-describe paradigm: it first analyzes the user request to identify core requirements and resolve ambiguities, then produces detailed guidance that enables faithful, controllable generation. We train ReaDe via a two-stage optimization: (i) reasoning-augmented supervision imparts analytic parsing with stepwise traces and dense captions; (ii) a multi-dimensional reward assigner enables stable, feedback-driven refinement for natural-style captions. Experiments across single- and multi-condition scenarios show consistent gains in instruction fidelity, caption accuracy, and downstream video quality, with strong generalization to reasoning-intensive and unseen inputs. ReaDe offers a practical route to aligning controllable video generation with accurately interpreted user intent.

## 1 Introduction

Video is a fundamental medium that captures the dynamics of the real world, and the ability to generate diverse, long-horizon, and semantically coherent videos is a critical stepping stone toward more general intelligence. In recent years, Diffusion Transformers (DiT) (Peebles & Xie, 2023; Ju et al., 2025) have dramatically advanced both the fidelity and temporal consistency of generated videos, making video generation increasingly viable for production-level applications such as cinematic content creation (kua, 2024; run, 2025; sor, 2024) and world simulation (He et al., 2025; Qin et al., 2024). As the quality advancement of generated content, users have begun to expect more fine-grained control, leveraging conditions such as reference images (Wei et al., 2024), segmentations (Lin et al., 2024), sketches (Wang et al., 2023), depth maps (Lin et al., 2024), human poses (Zhong et al., 2024), and camera trajectories (He et al., 2024; Bai et al., 2025a), as well as their flexible composition, to achieve greater controllability and creative freedom. However, empirical evidence shows that standard user inputs often fail to elicit the faithfully compelling videos users actually want. This exposes a core bottleneck for the video generation community: *aligning controllable generation with an accurately interpreted user intent*.

Prior studies (Yang et al., 2024) have highlighted that detailed prompts, which explicitly specify the target video's objects, actions, spatial layouts, camera behavior, style, and other scene attributes, can substantially improve controllability and quality during training. Motivated by this finding, several efforts (Chen et al., 2024; Ju et al., 2024; Fan et al., 2025) have explored video re-captioning to construct high-quality, detailed captions. These intricate captions are then used as training prompts for contemporary high-fidelity generators (Zheng et al., 2024b; Yang et al., 2024). At inference time, however, human-written inputs are often short and ambiguous, resulting in a pronounced mismatch between training prompts and real-world inputs. Such a discrepancy results in a generated video that neither follows user intent nor achieves high quality. To mitigate this gap, subsequent studies therefore investigate prompt interpretation, i.e., translating raw user inputs into the detailed forms expected by downstream generators, thereby improving controllability and quality. For example, Prompt-A-Video (Ji et al., 2024) introduces an LLM-based textual prompt adaptation framework to tailor prompts to a specific video generator, while Any2Caption (Wu et al., 2025) extends interpreta-

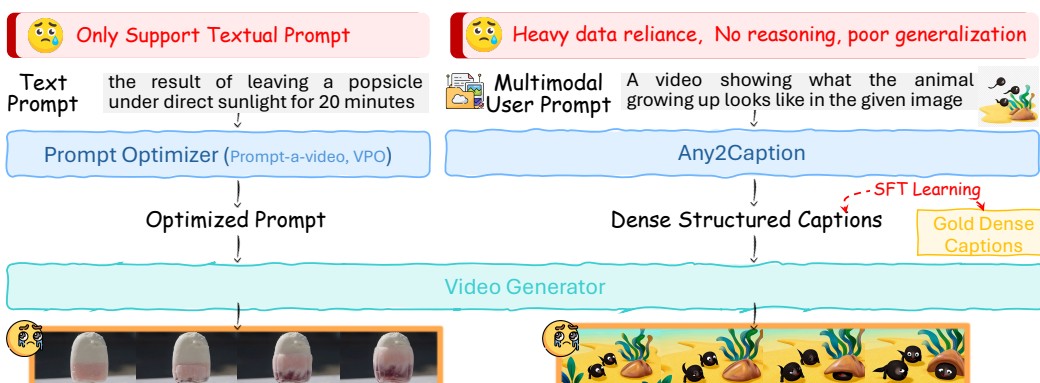

Figure 1: Text-only prompt optimizers and data-hungry multimodal methods (e.g., Any2Caption) remain brittle, performing poorly on reasoning-intensive and unseen instructions.

tion to multi-condition prompts. Unfortunately, existing approaches either address only textual conditions or depend on large amounts of condition-specific data, whose collection is often impractical. Moreover, supervised fine-tuning tends to encourage memorization, making it difficult to generalize to unseen or reasoning-intensive instructions that involve multiple cross-modal constraints.

To overcome the aforementioned limitations, we introduce **ReaDe**, a novel "reason-then-describe" instruction interpreter that is universal, generalizable, and model-agnostic, thereby enabling seamless integration with diverse downstream video generators to enhance controllability. Inspired by Chain-of-Thought (CoT) (Wei et al., 2022), **ReaDe** emulates a human-like reasoning process that systematically interprets the initial prompt into its core requirements, resolving cross-modal misalignments and ambiguities, and enriches it with explicit details to enable faithful and high-quality video generation. Technically, **ReaDe** instantiates a multimodal large language model backbone capable of ingesting textual and visual conditions, with extensions to camera and audio inputs. To effectively optimize **ReaDe**, we propose a multi-dimensional feedback reinforcement learning framework comprising two stages. In Stage 1, the interpreter is equipped with initial analytic parsing capabilities for instruction refinement, utilizing curated, reasoning-augmented data that pairs user inputs with stepwise reasoning traces and gold dense, detailed captions. In Stage 2, we design a multi-dimensional feedback reward assigner to overcome the intrinsic difficulty of evaluating naturally styled captions, enabling stable feedback-driven optimization that steers the model to infer user intent more accurately and generate detailed captions suitable for controllable video generation.

We conduct extensive experiments across diverse user instructions, including both single-condition and multi-condition settings. Results demonstrate that even when trained on a relatively small dataset, our proposed model produces more accurate and coherent captions, ultimately leading to higher-quality video generation. Further in-depth analysis reveals that the model exhibits strong comprehension of reasoning-driven instructions, consistently generating videos that align more closely with user intent. Comprehensive evaluations further confirm the robustness and generalization ability of our approach, showing competitive performance even in domains not directly covered during training. In summary, our contributions are threefold:

- We propose ReaDe, the first universal video instruction interpreter for controllable video generation, which employs a reason-then-describe paradigm inspired by CoT to refine various user inputs into detailed, faithful prompts across modalities.
- We design a multi-dimensional reward assigner that enables accurate assessment of generated caption quality while stabilizing and improving feedback-based refinement.
- We show that ReaDe consistently improves caption quality, video faithfulness, and cross-domain generalization across single-condition, multi-condition, and reasoning-driven instructions, highlighting the robustness and versatility of our framework.

## 2 RELATED WORK

Controllable video generation (Fang et al., 2024; He et al., 2024; Wei et al., 2024) has long become a hot topic in generative AI. Recent advances in DiT-based techniques (Peebles & Xie, 2023; Ju et al., 2025) have yielded models that can follow user-provided text prompts to produce high-quality, tem-

porally consistent, and photorealistic videos over extended durations. Early efforts primarily focused on text-controlled generation (Singer et al., 2022; Wu et al., 2023); as user expectations evolved, the community shifted toward providing frame-level control. To this end, fine-grained conditions—such as static images (Wang et al., 2024b; Guo et al., 2024; Zhang et al., 2023a), sketches (Zhao et al., 2023; Liu et al., 2025), human poses (Zhong et al., 2024; Ma et al., 2024; Karras et al., 2023), camera views (Zheng et al., 2024a; He et al., 2024), and even extra videos (Kara et al., 2024; Henschel et al., 2025), have been explored to enable precise, controllable video synthesis. Beyond single-condition control, multi-condition composition (Lin et al., 2024; Wang et al., 2023) has also been investigated. However, most existing methods are implemented as model-specific improvements, limiting their benefits to particular generators. This work instead aims to develop a universal, model-agnostic approach that can consistently enhance a variety of downstream video generators.

Despite impressive progress, the quality and accuracy of generative outputs remain highly dependent on a user's ability to craft precise, detailed prompts. A persistent challenge is the reliable interpretation of user inputs. In text-to-image generation, numerous studies (Mo et al., 2024; Li et al., 2024; Zhang et al., 2024; Wang et al., 2025) have explored prompt rewriting techniques that automatically enrich an initial prompt to provide more explicit guidance to the model. This issue is even more pronounced in text-to-video generation, where training commonly relies on detailed prompts while real-world user inputs tend to be concise and ambiguous, creating a distribution shift that degrades video quality Chen et al. (2024); Ju et al. (2024). Early work on video instruction enhancement, therefore, sought to optimize prompts for higher-fidelity video generation (e.g., Ji et al. (2024); Cheng et al. (2025); Gao et al. (2025), focusing solely on textual prompts). More recent efforts (Wu et al., 2025) have extended this to multiple conditions. Nonetheless, these approaches either remain text-only or rely on direct supervised fine-tuning, lacking explicit reasoning capabilities; as a result, they struggle with unseen conditions and reasoning-driven instructions. In contrast, our method emulates human-like interpretation via a reason-then-describe procedure and leverages fine-grained, feedback-driven optimization, enabling the model to autonomously learn how to interpret user instructions and thereby achieve stronger prompt enhancement and generalization.

## 3 METHODOLOGY

In this section, we present the proposed instruction interpreter (**ReaDe**) in detail. As illustrated in Fig. 2, **ReaDe** is built upon an existing multimodal large language model (Xu et al., 2025) of ingesting textual, visual, and audio conditions, further augmented with a camera encoder following Wu et al. (2025) to enable versatile condition interpretation. ReaDe is trained to follow a reason–then–describe paradigm through a two-stage pipeline. In Stage 1, we optimize the interpreter with supervised fine-tuning to impart stepwise reasoning capabilities. In Stage 2, we refine the model using multi-dimensional reward feedback. A dedicated reward assigner supplies fine-grained, content-aware signals along multiple aspects, guiding ReaDe to produce well-structured, detailed video captions that are maximally useful for downstream controllable video generation. After this two-stage training, ReaDe emerges as a universal, video-generator-agnostic instruction interpreter. Moreover, ReaDe can be further optimized by incorporating quality feedback from downstream video generators. Leveraging such feedback allows the interpreter to adapt in a generator-aware manner, thereby further improving overall video quality. We empirically validate this extended optimization strategy in our in-depth experiments (Sec. §4.4).

### 3.1 STAGE-1: COT-GUIDED REASONING INITIALIZATION

Drawing inspiration from CoT technique (Zhang et al., 2023b; Xu et al., 2024; Yao et al., 2024; Thawakar et al., 2025), which decomposes complex tasks into a sequence of manageable sub-problems, we design a four-step reasoning strategy tailored for generating dense, structured captions to facilitate controllable video generation (Fig. 2). Specifically, we first construct a CoT-style dataset and then perform preliminary fine-tuning to initialize the model's reasoning capability.

**CoT Data Construction.** We formalize the reasoning process into four steps, as outlined below:

- **Step-1: Interpretation of Textual Intent.** We first prompt GPT-4o (Hurst et al., 2024) to extract the user's core objective, determining whether the instruction involves *creation*, *addition*, or *modification* of elements, and identifying the specific content to be incorporated, particularly for complex generative requirements.

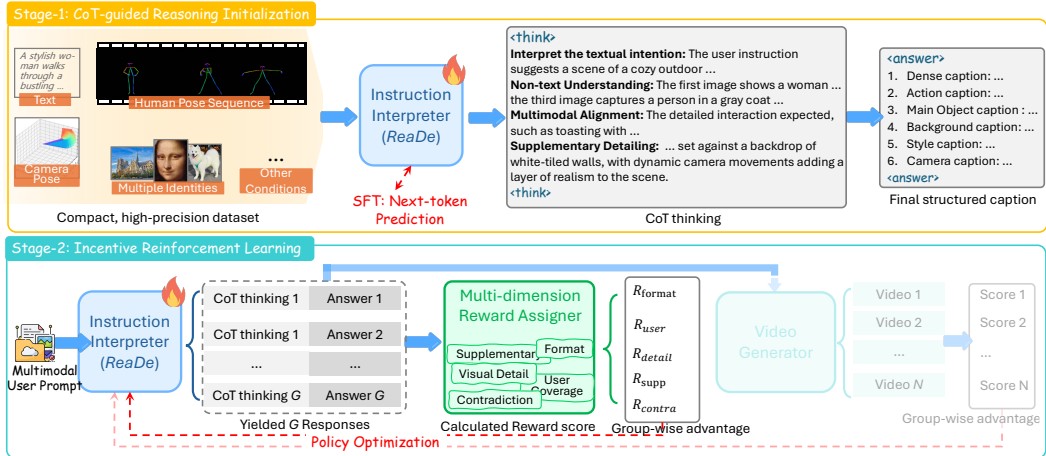

Figure 2: Overview of the training framework for the Instruction Interpreter (**ReaDe**). (1) CoT-guided reasoning initialization via supervised fine-tuning on instruction–thinking–answer triples, and (2) reinforcement learning with a multi-dimensional reward assigner and optional video-quality feedback from a frozen video generator.

- **Step-2: Non-Textual Understanding.** To capture details beyond text, we parse auxiliary modalities such as identities, human poses, or camera motions. These inputs are converted into descriptive cues using dedicated *X-to-Caption* models (Hurst et al., 2024; Bai et al., 2025b; Chai et al., 2025).
- **Step-3: Multimodal Alignment.** GPT-4o (Hurst et al., 2024) is then employed to align textual and non-textual instructions, ensuring consistency across modalities and producing an integrated interpretation of user intent.
- **Step-4: Supplementary Detail Completion.** Beyond users' inputs, certain contextual details must be inferred to achieve coherent video generation via GPT-4o (Hurst et al., 2024), including camera motions, missing environmental attributes, or stylistic elements.

Finally, the reasoning chain culminates in a dense, structured caption, adapted from Any2Caption (Wu et al., 2025), that consolidates all modalities into a unified and fine-grained description of the target video. To emulate the "thinking" style introduced in DeepSeek-R1 (Guo et al., 2025), we explicitly wrap intermediate reasoning steps with a `<think></think>` tag and encapsulate the final structured caption within an `<answer></answer>` tag. Complete examples of the prompts and reasoning traces for each step are provided in the Appendix §D.1. After deduplication and removal of overlaps, while ensuring diversity across instruction types, we curate a final dataset of 8.4K training examples. The concrete statistics are shown in Appendix §D.3. This high-quality, format-consistent corpus serves as the foundation for cold-start training, equipping the interpreter with an initial capacity for structured multimodal reasoning.

**SFT-based Optimization.** At this stage, we supervise the fine-tuning of the model based on the constructed dataset $D_{\text{cot}} = \{x_i, y_i\}_{i=1}^{|D_{\text{cot}}|}$. The optimization objective is:

$$\mathcal{L}_{\text{cot}} = -\mathbb{E}_{(x,y) \sim D_{\text{cot}}} \Big[ \sum_{t=1}^{|y|} \log \pi_\theta(y_t | x, y_{t-1}) \Big]. \tag{1}$$

## 3.2 STAGE-2: INCENTIVE REINFORCEMENT LEARNING

Following the initial reasoning-tuning stage, the model acquires a preliminary ability to perform step-by-step reasoning. However, our ultimate goal is to endow the model with reasoning skills that are not merely memorized from training data but can generalize robustly across diverse, real-world scenarios. To this end, we adopt a feedback-guided reinforcement learning framework to further refine the interpreter. Unlike mathematical problem solving, where rule-based and verifiable criteria can be directly applied, generating free-form video captions poses inherent challenges due to their open-ended nature. To address this, we incorporate two complementary reward signals. In addition to a basic *answer-format reward*, we design a *multi-dimensional content reward* that provides fine-

grained evaluations of generated captions across multiple aspects, implicitly capturing the quality of the underlying reasoning process. These rewards are aggregated in a group-wise manner (Guo et al., 2025; Meng et al., 2025) to ensure stable policy optimization.

### 3.2.1 REWARD ASSIGNMENT.

**Answer Structure Reward.** We encourage the model to follow the desired *thinking-then-answer* paradigm. In addition, the `<answer>` section is required to produce a six-part structured caption, for which adherence is likewise incorporated into the format reward.

$$R_{\text{format}} = \begin{cases} 1, & \text{if both \texttt{think} and \texttt{answer} formats are satisfied} \\ 0.2, & \text{if only one format is satisfied} \\ 0, & \text{if neither format is satisfied.} \end{cases} \tag{2}$$

**Multi-dimensional Content Reward.** To assess content quality beyond structural compliance, we design a multi-dimensional reward that reflects both user intent and the implicit reasoning process. Specifically, the gold caption is decomposed into three key aspects: **(1)** essential details explicitly required by the user textual instruction ($U$), **(2)** supplementary information derived from non-textual conditions ($S$), and **(3)** reasonable imaginative details that enhance coherence and realism ($Z$). For efficiency, we first conduct an offline extraction of gold dense structured captions to obtain the core elements associated with each aspect, including *objects*, *attributes*, *actions*, *style*, and *camera movements*. During training, we employ Qwen3-30B-A3B-Instruct-2507 (Yang et al., 2025) as a judging model to evaluate the predicted caption's $\hat{y}$ coverage of these elements:

$$R_{\text{user}} = \frac{1}{|U|} \sum_{u \in U} \mathbf{1}[\text{M}(u, \hat{y})]; \quad R_{\text{detail}} = \frac{1}{|S|} \sum_{s \in S} \mathbf{1}[\text{M}(s, \hat{y})]; \quad R_{\text{supp}} = \frac{1}{|Z|} \sum_{z \in Z} \mathbf{1}[\text{M}(z, \hat{y})], \tag{3}$$

where $\text{M}(\cdot)$ is the matching function to determine if the predicted caption contained the original.

Furthermore, we observe that longer outputs often introduce contradictions or internal inconsistencies. To mitigate this, we incorporate an additional consistency factor $R_{\text{contra}} = \mathbf{1}[\text{contradict}(\hat{y})]$. The final content reward is computed as a weighted aggregation of the above components, ensuring a balanced signal that promotes faithful, detailed, and coherent reasoning:

$$R = \alpha R_{\text{user}} + \rho R_{\text{detail}} + \gamma R_{\text{supp}} - \lambda R_{\text{contra}}, \tag{4}$$

where $\alpha, \rho, \gamma,$ and $\lambda$ are the hyper-parameters.

### 3.2.2 DATA CONSTRUCTION FOR RL LEARNING

To effectively reward the model's reasoning process, we curate a dedicated dataset that incorporates both explicit reasoning traces and offline–extracted core elements, as described above. Instead of randomly sampling from existing instances, we design a data selection pipeline to ensure that the final training set satisfies two key criteria: (i) balanced complexity of user instructions, achieved by controlling for textual types and the number of non-textual conditions, and (ii) the availability of detailed structured captions paired with the corresponding core elements required for reward computation. The complete curation pipeline is provided in Appendix §D.2. Through this process, we obtain a total of 8.3K training examples spanning diverse settings, including multiple identities, depth maps, camera motions, and human poses.

### 3.2.3 OPTIMIZATION VIA GRPO

Reinforcement learning has recently emerged as a dominant paradigm for eliciting reasoning capabilities in large models. In particular, Group Relative Policy Optimization (GRPO) (Guo et al., 2025), a variant of PPO (Schulman et al., 2017), eliminates the dependency on a critic model, thereby reducing training costs by directly comparing groups of sampled responses. This makes GRPO especially suitable for reasoning-intensive tasks. In our framework, we also leverage GRPO to optimize ReaDe. Given an input instruction $q$, the policy generates $G$ distinct candidate responses $O = \{o_1, \ldots, o_G\}$ through sampling. Each response $o_i$ is assigned a reward $R_i$ as described in the previous section. To stabilize optimization, GRPO computes a group-relative advantage by normalizing rewards with respect to the group mean and standard deviation:

$$A_i = \frac{R_i - \text{mean}(\{R_j\}_{j=1}^{G})}{\text{std}(\{R_j\}_{j=1}^{G})}, \tag{5}$$

Table 1: Comparison of *Multiple Identities* controlled video generation performance.

| Model | CLIP-T↑ | DINO-I↑ | Smoo.↑ | Aest.↑ |
|---|---|---|---|---|
| ConceptMaster | 16.04 | 36.37 | 94.71 | 5.21 |
| Any2Caption | 17.15 | 39.42 | 95.05 | 5.48 |
| ReaDe | **18.64** | **45.28** | **95.36** | **5.59** |

Table 2: Comparison of *Depth* controlled video generation performance.

| Model | CLIP-T↑ | MAE↓ | Smoo.↑ | Aest.↑ |
|---|---|---|---|---|
| Ctrl-Adapter | 20.37 | 25.63 | 94.53 | 4.63 |
| Any2Caption | 23.30 | 21.87 | 95.54 | 5.31 |
| ReaDe | **24.16** | **18.79** | **95.56** | **5.75** |

Table 3: Comparison of *Camera* controlled video generation performance.

| Model | CLIP-T↑ | RotErr↓ | TransErr↓ | Smoo.↑ |
|---|---|---|---|---|
| MotionCtrl | 19.67 | 1.54 | 4.49 | 96.13 |
| Any2Caption | 20.16 | 1.45 | 4.37 | 96.16 |
| ReaDe | **21.57** | **1.30** | **3.46** | **96.37** |

Table 4: Comparison of *Human Pose* controlled video generation performance.

| Model | CLIP-T↑ | PAcc.↑ | Smoo.↑ | Aest.↑ |
|---|---|---|---|---|
| FollowYourPose | 21.11 | 30.47 | 91.71 | 4.95 |
| ReaDe | **22.45** | **32.76** | **93.14** | **5.86** |

This group-relative advantage $A_i$ encourages the model to prioritize responses with higher relative quality. To prevent the optimized policy $\pi_\theta$ from diverging excessively from the reference model $\pi_{\text{ref}}$, a KL-divergence regularization term is introduced. The overall optimization objective is:

$$\max_{\pi_\theta} \mathbb{E}_{[q \sim D_{\text{rl}}, \{o_i\}_{i=1}^G \sim \pi_\theta(O|q)]}$$

$$\left[ \frac{1}{G} \sum_{i=1}^{G} \min\left( \frac{\pi_\theta(o_i)}{\pi_{\theta_{\text{old}}}(o_i)} A_i, \text{clip}\left( \frac{\pi_\theta(o_i)}{\pi_{\theta_{\text{old}}}(o_i)}, 1-\epsilon, 1+\epsilon \right) A_i \right) - \beta \, \mathbb{D}_{\text{KL}}\left( \pi_\theta \| \pi_{\text{ref}} \right) \right], \tag{6}$$

where $\beta$ is a regularization coefficient that balances optimization efficiency with stability by constraining deviations from the reference policy.

## 4 EXPERIMENTS

### 4.1 SETTINGS

Our instruction interpreter is initialized from Qwen2.5-Omni (Xu et al., 2025) and augmented with an external camera encoder (Wu et al., 2025) to compensate for its lack of understanding of camera motion. We first conduct a lightweight camera-to-text pretraining that translates camera signals into textual cues, which are then consumed by the interpreter. At stage 1, we curate four condition types, including multi-identity references, camera motion, depth maps, and human pose, yielding 8.4K training instances with both straightforward and edit-style prompts. Training uses an initial learning rate of 1e-5 with a cosine scheduler. At stage 2, we construct the training data, comprising a total of 8.3K instances. We implement reinforcement learning using a constant learning rate of 2.5e-6, 8 rollouts per prompt, and a KL coefficient of 0.001. We evaluate on the dataset proposed in FullDiT Ju et al. (2025), reporting performance under both single- and combined-condition controls without specification. For a more detailed implementation, refer to Appendix §4.4.

### 4.2 MAIN RESULTS

We compare three prompt regimes for multiple downstream controlled video generators: the original short prompt, the structured caption produced by Any2Caption, and our interpreter. Across single-condition controls, our captions consistently yield higher alignment and quality, as shown in Tables 1, 2, 3, 4. We attribute the gains to more accurate parsing of user intent and fewer cross-modal inconsistencies in the generated descriptions. As demonstrated in Table 5, the improvements become more pronounced under multi-condition composition (e.g., Camera+Depth+IDs), where naïve SFT struggles to reconcile conflicting constraints and alignment.

### 4.3 ABLATION STUDY

In this section, we analyze the necessity of our two-stage learning strategy and the contributions of different reward designs.

**The Impact of Reward Aspects.** Applying rewards alone (e.g., only $R_{\text{user}}$) yields weaker performance, underscoring the importance of explicit reasoning for learning structured intention understanding. Among the reward components, $R_{\text{user}}$ and $R_{\text{detail}}$ contribute the most: their combination

Table 5: Quantitative comparison of generation performance under compositional conditions. `C`, `D`, and `I` denote `camera`, `depth` and `multiple identities` conditions, respectively.

| Cond. | Method | Text | Camera | | Identities | | Depth | Overall Quality | | |
|---|---|---|---|---|---|---|---|---|---|---|
| | | CLIP-T↑ | RotErr↓ | TransErr↓ | DINO-I↑ | CLIP-I↑ | MAE↓ | Smoothness↑ | Dynamic↑ | Aesthetic↑ |
| C+I | FullDiT | 14.81 | 1.37 | 4.04 | 25.63 | 64.14 | - | 94.43 | 28.87 | 4.99 |
| | Any2Caption | 19.03 | **1.30** | 4.36 | 26.75 | 68.45 | - | 94.38 | 34.99 | 5.25 |
| | ReaDe | **19.35** | 1.34 | **4.01** | **28.45** | **69.73** | - | **95.14** | **35.12** | **5.26** |
| C+D | FullDiT | 20.80 | 1.57 | 3.88 | - | - | 32.15 | 95.36 | 30.12 | 4.82 |
| | Any2Caption | 21.19 | 1.49 | 4.41 | - | - | 25.37 | 95.40 | 30.10 | 4.96 |
| | ReaDe | **21.35** | **1.40** | **3.54** | - | - | **25.34** | **95.79** | **32.47** | **5.01** |
| D+I | FullDiT | 20.01 | - | - | 35.24 | 57.82 | **23.00** | 93.15 | 32.21 | 4.96 |
| | Any2Caption | 20.76 | - | - | 36.25 | 63.48 | 24.78 | 92.50 | **36.43** | **5.18** |
| | ReaDe | **23.14** | - | - | **37.89** | **64.21** | 23.08 | **93.41** | 35.48 | 5.01 |
| C+D+I | FullDiT | 18.49 | 2.05 | 7.74 | 35.86 | 64.25 | 18.37 | 92.02 | 30.09 | 3.91 |
| | Any2Caption | 19.52 | 1.57 | 7.74 | 38.74 | 64.37 | 17.41 | 93.03 | 32.81 | 4.99 |
| | ReaDe | **21.24** | **1.34** | **5.28** | **39.46** | **66.17** | **17.03** | **95.04** | **33.47** | **5.21** |

Table 6: Ablation study on multiple identity reference images-conditioned video generation.

| CoT | Reward | | | | Intension Reasoning | | Identities | | Video Quality | |
|---|---|---|---|---|---|---|---|---|---|---|
| | $R_{user}$ | $R_{detail}$ | $R_{supp}$ | $R_{contra}$ | Acc↑ | Qual.↑ | CLIP-I↑ | DINO-I↑ | Smoothness↑ | Dynamic↑ |
| ✓ | - | - | - | - | 62.41 | 3.12 | 16.28 | 39.74 | 92.15 | 4.86 |
| - | ✓ | - | - | - | 58.32 | 2.87 | 14.92 | 37.63 | 90.84 | 4.71 |
| - | ✓ | ✓ | - | - | 66.75 | 3.44 | 17.53 | 42.11 | 93.26 | 5.02 |
| - | ✓ | ✓ | ✓ | - | 68.93 | 3.57 | 18.02 | 43.27 | 94.12 | 5.21 |
| - | ✓ | ✓ | ✓ | ✓ | 70.26 | 3.62 | 18.25 | 44.01 | 94.57 | 5.33 |
| ✓ | ✓ | ✓ | ✓ | ✓ | 73.45 | 3.79 | 18.64 | 45.28 | 95.36 | 5.59 |

Figure 3: Generalization capability of **ReaDe**. The heatmap shows the dense caption intention accuracy under different training–evaluation condition pairs. The y-axis corresponds to training conditions, while the x-axis denotes evaluation conditions.

significantly boosts reasoning accuracy and video quality. Additional rewards, including $R_{supp}$ for supplementary details and $R_{contra}$ for consistency, provide further complementary gains, especially in mitigating contradictions and enhancing smoothness.

**The Impact of Learning Strategy.** We further compare three training strategies: (i) CoT-only, (ii) GRPO-only, and (iii) the proposed combination. As shown in Table 6, CoT initialization alone establishes a strong baseline, improving reasoning accuracy and identity preservation over short-prompt baselines even without explicit reward supervision. In contrast, GRPO without CoT suffers from unstable optimization; however, with carefully designed rewards, it can partially compensate for the lack of reasoning priors. By combining the two, our method inherits the stability of CoT-based reasoning and the adaptability of GRPO-based optimization, achieving the best overall performance across reasoning accuracy, identity fidelity, and video quality metrics.

## 4.4 IN-DEPTH ANALYSES

**Generalization Capability of Different Conditions Types.** We further investigate the generalization ability of the proposed model under different condition types. Specifically, the model is trained on one condition and evaluated on the others, with the intention accuracy score computed

Table 7: Performance of different prompting optimization strategies on VBench (Huang et al., 2024). The video backbone is CogVideoX-2B (Yang et al., 2024). SC: Subject Consistency, BC: Background Consistency, MS: Motion Smoothness, DD: Dynamic Degree, MO: Multiple Objects, AS: Appearance Style, S: Scene. The best results are in **bold**, and the second-best results are underlined.

| Model | SC | BC | MS | DD | MO | AS | S |
|---|---|---|---|---|---|---|---|
| Original prompts | 94.60 | 95.90 | 97.60 | 60.00 | 40.17 | 22.60 | 28.34 |
| Promptist (Hao et al., 2023) | 95.80 | 96.60 | 98.40 | 56.00 | 27.44 | 23.12 | 18.37 |
| GLM-4 (GLM et al., 2024) | 95.10 | 96.30 | 98.20 | 60.00 | 68.40 | 23.47 | 55.51 |
| Prompt-A-Video (Ji et al., 2024) | 95.30 | 95.90 | 98.30 | 54.00 | 68.26 | 22.33 | 43.85 |
| VPO (Cheng et al., 2025) | - | - | - | - | 71.17 | 24.20 | 55.83 |
| Any2Caption (Wu et al., 2025) | 95.50 | 96.10 | 98.10 | 58.00 | 69.85 | 22.80 | 49.12 |
| ReaDe (SFT) | 95.73 | 96.34 | 98.27 | 59.36 | 68.75 | 23.05 | 54.36 |
| ReaDe (SFT+GRPO) | 96.17 | 96.71 | **98.45** | 61.34 | 70.54 | 24.10 | 55.78 |
| ReaDe (w/ Video Generator) | **96.41** | **98.44** | 98.01 | **65.92** | **71.89** | **26.34** | **56.78** |

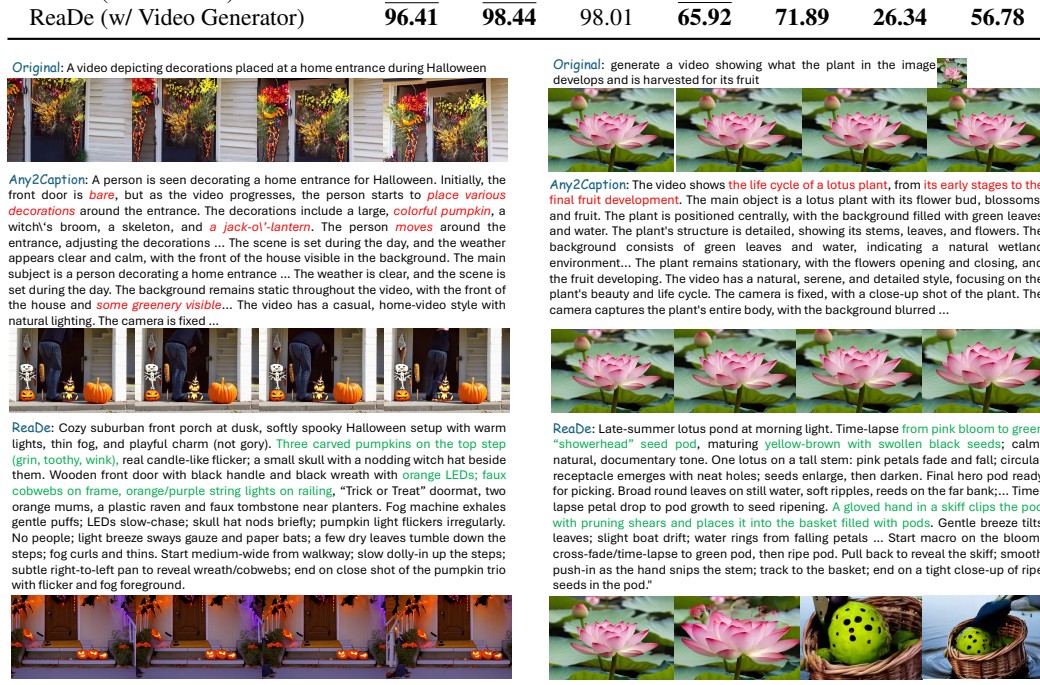

Figure 4: Illustration of prompt optimization for raw prompts. The left panel shows text-to-image generation results produced by CogVideoX-2B, while the right panel presents image-to-text generation results obtained with CogVideoX-5B-I2V. Some prompts are omitted due to space constraints.

under the overall description setting. The results are presented in Fig. 3. We observe that the degree of generalization varies across condition types. For example, training on depth demonstrates relatively strong transferability to other conditions, which can be attributed to the fact that depth provides dense signals that implicitly encode information relevant to other modalities, such as images or camera parameters. Moreover, our method consistently exhibits generalization ability across all conditions and achieves comparable performance regardless of the training condition.

**Comparison of Prompting Optimization Strategies.** We compare different prompting optimization strategies for enhancing prompt quality, with results reported on the VBench benchmark (Table 7). Compared with directly using raw prompts, all optimization strategies lead to consistent improvements in generation quality. In particular, our proposed ReaDe, trained with a two-stage learning scheme, achieves superior performance across multiple dimensions compared to Any2Caption, demonstrating its ability to learn more informative and effective dense captions. Furthermore, when compared with Prompt-A-Video and VPO, our method attains comparable or better generation quality through its universal instruction interpreter, indicating that the captions produced by our prompt generator are both semantically coherent and aligned with user intent. Moreover, by incorporating feedback signals from downstream video generators, our model further refines its outputs and achieves the best overall results.

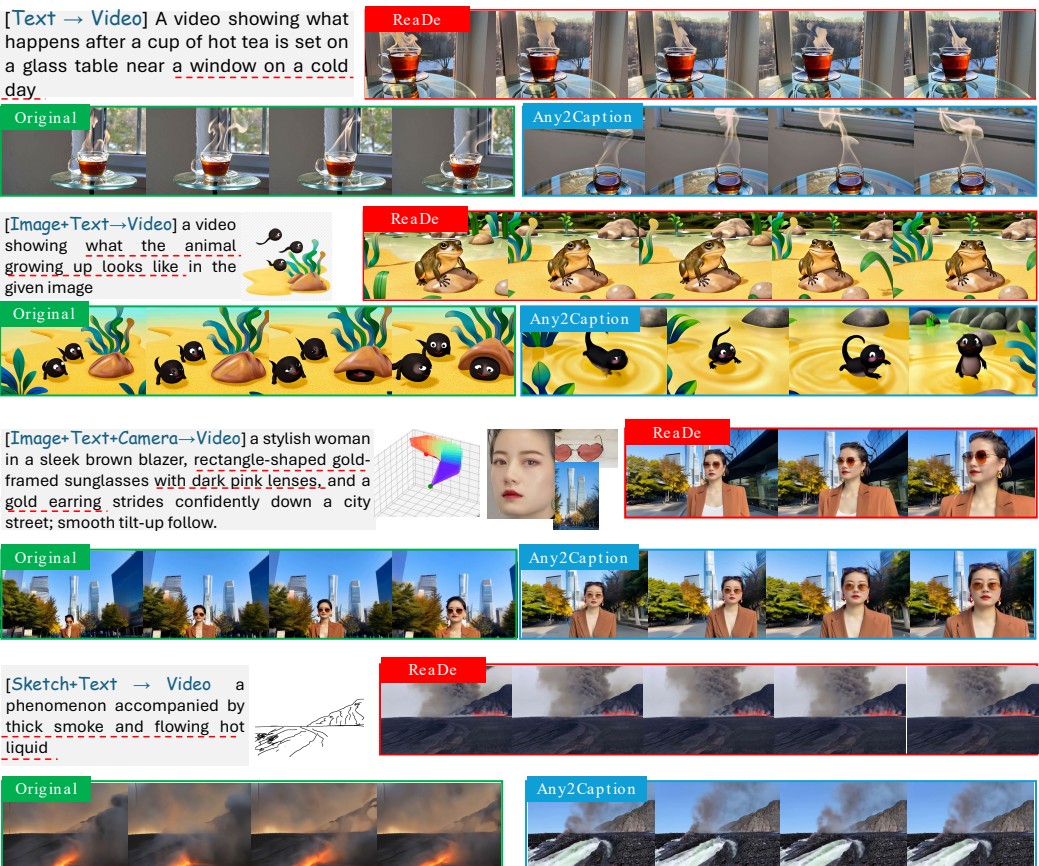

Figure 5: Qualitative comparison of the generation quality across original prompts, interpreted prompts by Any2Caption and ReaDe. The first two rows are generated via Kling1.6, the third is generated via FullDiT, and the last one is generated with SketchVideo.

**Case Study on Reason-intensive Prompts.** We further conduct a qualitative case study on reason-intensive prompts, with representative results illustrated in Fig. 5. It can be observed that ReaDe generates outputs that more faithfully align with user intent while maintaining higher temporal coherence. For example, in text-to-video generation, our model not only captures the rising steam but also depicts the snow outside the window, thereby rendering a vivid scene of a cold day. In image-to-video generation, ReaDe demonstrates reasoning ability by recognizing that the animal in the input image matures into a frog and accordingly generates a faithful video, better satisfying user expectations. Moreover, in scenarios involving multi-condition combinations, our method produces more coherent and faithful results. In addition, we compare the intermediate prompts produced by Any2Caption and by our interpreter (Fig. 4). We find that Any2Caption often lacks sufficient reasoning ability, resulting in exaggerated or superficial descriptions, whereas our approach yields more precise and faithful prompts that effectively guide video generation.

## 5 CONCLUSION

In this work, we introduced **ReaDe**, a universal instruction interpreter that plugs into downstream video generators and translates heterogeneous conditions into faithful, generator-ready prompts. Built upon an MLLM, ReaDe ingests diverse multimodal conditions for interpretation. To train ReaDe, we adopt a two-stage learning paradigm: (i) supervised fine-tuning to instill stepwise reasoning, followed by (ii) multi-dimensional reward feedback to optimize prompt quality and alignment. Extensive experiments on text-only and single-/multi-condition controlled video generation demonstrate consistent gains in faithfulness, temporal coherence, and controllability, indicating that ReaDe produces prompts that better match user intent and the requirements of modern video generators. Looking ahead, we plan to extend ReaDe to broader modalities and longer-horizon reasoning, and to explore human-in-the-loop and safety-aware optimization for real-world deployment.

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

APPENDIX INDEX

This supplementary material includes the following sections:

## A  THE USE OF LARGE LANGUAGE MODELS (LLMs)

In this paper, we employ the large language models (LLMs) for dataset construction and clarity and readability improvement of paper writing. Specifically, we adopt the GPT-4o to construct the training dataset. Moreover, we employ Qwen3-30B as the reward model to evaluate the prediction quality for model optimization. Furthermore, LLMs are employed to polish sentence structure, correct grammatical errors, and enhance the overall presentation of our draft. The technical content, research ideas, experimental design, analysis, and conclusions were entirely conceived, implemented, and validated by the authors without reliance on LLMs.

## B  ETHICS STATEMENT

All training data used in this work are non-public but authorized for research under existing licenses and confidentiality agreements. The data contain no personally identifiable information (PII) or sensitive personal attributes, and no third-party intellectual property is used without permission. To construct auxiliary training material, we employed both open- and closed-source generation models; all generated samples were manually screened to reduce risks of harmful content, discrimination, or bias.

Our approach is implemented on an open-source foundation model. While this choice helps mitigate safety and fairness concerns, no generative system can fully eliminate the possibility of unintended or harmful outputs. We therefore caution that downstream deployments—especially in sensitive domains—should incorporate appropriate safeguards, including content moderation, safety filtering, and bias assessments.

For evaluation, we report results exclusively on publicly available benchmarks to ensure transparency and comparability with prior work. This study involves no human-subject experiments and does not process PII. The work adheres to community standards on lawful data use, privacy protection, and research integrity. Our contributions are intended solely for academic and scientific exploration, and we explicitly discourage any misuse of the methods described.

## C  REPRODUCIBILITY STATEMENT

To ensure the reproducibility of our work, we have made a concerted effort to provide all necessary details and materials. We provide comprehensive details of the proposed ReaDe framework, including its definition, input–output formulation, and implementation (Section §3). The model backbone and training methodology are described in detail in Section §3. Appendix §D.4 provides in-depth analyses of the rationality of the design of the proposed method. We further report all hyperparameter settings and training configurations in Section §4.1 and Appendix §D.3, using fixed random seeds to ensure the replicability of the experiments. We provide detailed prompts, along with the amount of data used at each training stage, which are thoroughly documented in Appendices §D.1 and §D.2. Finally, we will release the full codebase and data processing scripts to the community upon acceptance.

# D EXTENDED EXPERIMENTAL SETTINGS

## D.1 DATA CONSTRUCTION FOR CoT

Here, we provide a detailed description of the data construction pipeline for the CoT dataset used in Stage 1 of our training framework. Our dataset source is derived from Wu et al. (2025). The goal of this dataset is to encourage models to explicitly reason through user instructions before producing the final interpreted intent.

**Step-1: Interpretation of Textual Intent.** In the first step, we employ GPT-4o to interpret the intention expressed in the users' textual instructions. Specifically, the model is required to identify whether the user requests a new video to be generated or an operation (e.g., modification, addition, or deletion of content) to be performed on an existing video. A prompt template example is shown below, using the case of multiple-identity-controlled video generation.

---

Step-1: Interpretation of Textual Intent (condition example: Multiple Identities)

You are a reasoning agent. Your task is to infer the user's intention based on the instructions provided for generating or editing a video. Your specific task is to \*\*interpret the user's intent\*\* by following these steps:
- Determine whether the user is asking to generate a new video or perform operations such as modifying, adding, or removing certain content.
- Identify the core objective or thematic focus of the instruction.
- If the instruction is explicit, provide a direct interpretation. - If the instruction requires reasoning, analyze it step by step before giving your final interpretation.
\*\*Output Requirements:\*\*
- Always provide your response in clear, complete sentences describing the user's actual intent.
- Do not include irrelevant explanations or extra commentary. - Strictly follow the output format shown below.

=====================================================================
PLEASE     STRICTLY     FOLLOW     THE     OUTPUT     FORMAT.
=====================================================================
Input instruction: %
Output:

---

**Step-2: Non-Textual Understanding.** In this step, we derive detailed semantic descriptions from the user-provided *non-textual conditions*. For each condition type, we convert signals into compact, sentence-level captions that capture key *objects*, *attributes*, and *relations* (and, when applicable, motion and geometry). We adopt condition-specific "condition-to-caption" tools as follows:

- **Image / Identity reference images.** For single-frame visual references (including identity reference images), we employ **GPT-4o** (Hurst et al., 2024) to produce dense descriptions that emphasize identity-defining attributes (e.g., hairstyle, clothing, accessories), salient objects, and their relationships. The output is a concise yet attribute-rich caption suitable for conditioning downstream generation.
- **Depth and human-pose sequences.** For geometric or kinematic conditions (depth maps and human-pose sequences), we adopt an off-the-shelf video captioner (e.g., **Tarsier** (Yuan et al., 2025)) to generate temporally grounded descriptions of the scene content. The captions focus on spatial layout (occlusions, support relations), motion movements, and object interaction, aligning them with the provided depth/pose cues.
- **Camera trajectory.** As dedicated camera-captioning tools are not yet available, we apply an automatic video captioner (e.g., **AuroCap** (Chai et al., 2025)) to the original video aligned with the estimated camera trajectory, then *filter and retain only camera-related clauses* (e.g., pan left/right, tilt up/down, dolly in/out, zoom in/out). This yields a concise description of camera motion divorced from scene semantics.

**Step-3: Multimodal Alignment.**    In this step, we infer the aligning information across *multiple modalities* when the user provides both textual instructions and non-textual conditions. The goal is to analyze the coherence, complementarity, and potential conflicts among these heterogeneous inputs. In particular:

- When conflicts arise, we prioritize the explicit requirements specified in the textual instruction, as these most directly reflect the user's primary intent.

- When the non-textual conditions provide additional details that do not conflict with the text, we incorporate them to enrich the final representation.

- When multiple instructions or references are provided, we assess their interrelations, highlighting consistency or resolving contradictions to produce a unified multimodal intent.

To operationalize this step, we prompt GPT-4o with a structured template. An illustrative example is shown below, using the case of video generation with multiple identities.

---

Step-3: Multimodal Alignment (condition example: Multiple Identities)

You are a reasoning agent. Your task is to analyze the alignment and potential conflicts between the **textual instruction** and the **non-textual description(s)** provided by the user. Specifically:
- Identify points of alignment across modalities.
- Detect any conflicts (e.g., attribute mismatches, incompatible actions, scene discrepancies).
- Resolve conflicts by *prioritizing the textual instruction*, while retaining compatible non-textual details.
- Produce a concise, unified interpretation of the final multimodal intent.

**Output Requirements:**
- Provide your response in clear, complete sentences describing the final alignment.
- Explicitly state both consistencies and conflicts (if any), followed by the resolved interpretation.
- Do not include irrelevant explanations or extra commentary.
- Strictly follow the output format shown below.
=======================================================================
PLEASE        STRICTLY        FOLLOW        THE        OUTPUT        FORMAT.
=======================================================================
Input textual instruction: %
Input non-textual description:%
Output:

---

**Step-4: Supplementary Detail Completion.**    The final step aims to enrich the multimodal intent with specific details that may not be explicitly provided by the user. For example, when the input consists of depth maps and textual instructions, the intended video style may remain unspecified. In such cases, the model is expected to infer and supplement plausible details (e.g., scene style, atmosphere) to produce a complete and coherent description. Importantly, this supplementation is not arbitrary. During construction, we leverage *gold dense video captions*, which offer detailed annotations for each component. The model is guided to ground its imaginative completion on these references, integrating them with the available inputs while clearly distinguishing inferred content. To achieve this, we employ GPT-4o to perform the reasoning and generate the final enriched descriptions.

Step-4: Supplementary Detail Completion (condition example: Multiple Identities)

You are a reasoning agent specialized in supplementing missing details for multimodal video generation. Your task is to enrich the current description with plausible details that are not explicitly mentioned, grounded in the reference gold video description.

You will be provided with two pieces of information:
1. A **gold video description**, which contains dense and detailed annotations.
2. The **current available description**, which may be incomplete.
Your specific job is to:
- Compare the gold description against the current description. - Identify the key details that are missing or not mentioned (e.g., scene style, background elements, temporal context, atmosphere).
- Provide only the missing details, written in clear, concise, and natural English sentences.
- The supplementation must remain consistent with the gold description and should not introduce arbitrary or contradictory information.

—

**Output Requirements:**
- Provide your response in clear, complete sentences describing only the missing supplementary information.
- Do not restate the existing description.
- Do not include irrelevant explanations or extra commentary.
- Strictly follow the output format shown below.
=====================================================================
PLEASE      STRICTLY      FOLLOW      THE      OUTPUT      FORMAT.
=====================================================================
Input gold video description: %
Input current available description:%
Output:

## D.2    DATA CONSTRUCTION FOR RL

The primary goal of this stage is to ensure both efficiency and feasibility in calculating the score in the reward model. To this end, we distill the information into several key aspects that align with the reasoning process. As illustrated in Table 8, we present a concrete example: the table shows the gold dense caption alongside the user's input prompt. Table 9 shows the corresponding extracted results, including the user's textual input, the key information supplemented from non-textual conditions, and the additional imaginary details.

## D.3    DETAILED IMPLEMENTATIONS

**Training Settings.**    As shown in Fig. 6, our model is a multimodal LLM initialized from Qwen2.5-Omni (Xu et al., 2025). Except for the camera encoder, all other components are directly inherited from Qwen2.5-Omni. Following Wu et al. (2025), the camera encoder adopts a vision-encoder architecture with an input channel of 96, a patch size of 16, a depth of 8, and 8 attention heads. To enable the model to effectively interpret camera information, we conduct camera-understanding pre-training in which only the camera encoder is updated while all other components remain frozen.

Table 10 summarizes the common hyperparameter configurations set in Stage 1/2. In Stage 2, we set the maximum completion length to 768 and the rollout number to 8 for each input. The weights for different reward components are set as $\alpha = 1.0$, $\rho = 1.0$, $\gamma = 0.8$, and $\lambda = 0.7$. The KL coefficient is $\beta = 0.001$, and the clipped policy-gradient loss is computed with $\epsilon = 0.2$.

**Training Dataset.**    As shown in Table 11, we report the number of instances for each condition used in Stage 1 and Stage 2 training. In constructing the training dataset, we also ensure diversity, for example, the number of identities ranges from 1 to 5, and the human-pose condition involves

Table 8: Example of gold dense caption and user's input textual instruction.

| Gold Dense Caption | 1. Overall description: A woman is seated at a desk in a minimalistic room, working on a laptop. She is wearing a grey camouflage-patterned shirt and has long blonde hair. Initially, she is focused on her laptop, occasionally glancing at a smartphone beside her. As time progresses, she shifts her attention to the smartphone, eventually picking it up and moving it out of the frame. The video concludes with the desk and wall in the background, now empty.2. Main object description: A young woman with long, straight blonde hair, light-colored eyes, and a fair complexion, in her 20s or 30s and of Caucasian ethnicity, is seated at a white desk. She is wearing a short-sleeved, gray camouflage-patterned t-shirt and a delicate gold necklace. Her build is slim, and she has a friendly and engaging demeanor, often smiling and making eye contact with the camera. She appears to be happy and enthusiastic as she speaks, occasionally glancing down at a laptop in front of her and a smartphone to her right.3. Background description: The background is a plain, light-colored wall, creating a minimalist and clean setting. The desk is white, and the overall lighting is bright and even, suggesting an indoor environment during the daytime.4. Movement description: A woman in a gray coat sat in front of the computer, talking, and then the woman left the camera.5. Style description: Clean, minimalist, and professional.6. Camera description: The camera is fixed. The camera is roughly at the same height as the person, maintaining a medium close-up shot of the upper body. As the person moves, the shot transitions from a frontal view to a profile view, with the person moving from the center of the frame to exiting the frame. |
|---|---|
| Textual Instruction | A woman in a minimalistic room sits at a white desk, working on her laptop. She wears a grey camouflage-patterned shirt with long blonde hair. Gradually, she shifts her focus from her laptop to the smartphone beside her, eventually picking it up and moving it out of view. The scene is well-lit with a plain wall in the background, conveying a clean and professional style. The camera remains fixed at her height, transitioning from a front to a profile view as she exits the frame. |

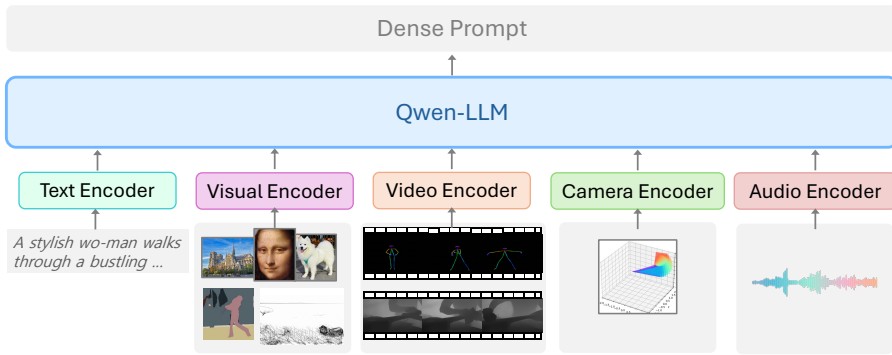

Figure 6: Illustration of the multimodal encoding framework. Various user-provided conditions are processed by their corresponding encoders (text, visual, video, audio, and camera), and the extracted features are integrated by the Qwen-LLM to perform reasoning. The model outputs a deeply interpreted dense prompt for the downstream video generator.

varying numbers of people. Our focus is not on collecting large quantities of data, but on ensuring high-quality training. Overall, we use only 16K instances, which is substantially smaller than the dataset in Wu et al. (2025) (337K).

Table 9: Example of an extracted JSON file containing the user input, supplementary information from non-textual conditions, and inferred imaginary details.

| | |
|---|---|
| User Input Key Info. | {'objects': [{'name': 'woman', 'attributes': ['sits', 'long blonde hair', 'grey camouflage-patterned shirt']}, {'name': 'room', 'attributes': ['minimalistic']}, {'name': 'desk', 'attributes': ['white']}, {'name': 'laptop', 'attributes': []}, {'name': 'smartphone', 'attributes': []}, {'name': 'wall', 'attributes': ['plain']}], 'actions': ['shifts focus from laptop to smartphone', 'picks up smartphone', 'moves smartphone out of view'], 'camera': ['fixed at her height', 'transitioning from front to profile view'], 'style': ['clean', 'professional']} |
| Supplementary key Info. | {'objects': [{'name': 'woman', 'attributes': ['slim build', 'friendly demeanor', 'smiling', 'making eye contact']}, {'name': 'laptop', 'attributes': []}, {'name': 'smartphone', 'attributes': []}, {'name': 'desk', 'attributes': ['white']}, {'name': 'wall', 'attributes': ['light-colored', 'minimalist', 'clean']}], 'actions': ['working on laptop', 'glancing at smartphone', 'talking'], 'camera': ['medium close-up shot'], 'style': ['clean', 'minimalist', 'professional']}","{'objects': [{'name': 'woman', 'attributes': ['sits', 'long blonde hair', 'grey camouflage-patterned shirt']}, {'name': 'room', 'attributes': ['minimalistic']}, {'name': 'desk', 'attributes': ['white']}, {'name': 'laptop', 'attributes': []}, {'name': 'smartphone', 'attributes': []}, {'name': 'wall', 'attributes': ['plain']}], 'actions': ['shifts focus from laptop to smartphone', 'picks up smartphone', 'moves smartphone out of view'], 'camera': ['fixed at her height', 'transitioning from front to profile view'], 'style': ['clean', 'professional']} |
| Imaging Key Info. | {'objects': [{'name': 'woman', 'attributes': ['seated', 'long blonde hair', 'wearing gray camouflage-patterned shirt', 'slim build', 'friendly demeanor', 'smiling', 'making eye contact', 'happy', 'enthusiastic']}, {'name': 'laptop', 'attributes': []}, {'name': 'smartphone', 'attributes': []}, {'name': 'desk', 'attributes': ['white']}, {'name': 'wall', 'attributes': ['light-colored', 'minimalist', 'clean']}], 'actions': ['working on laptop', 'glancing at smartphone', 'picking up smartphone', 'moving smartphone out of frame', 'talking', 'leaving camera'], 'camera': ['fixed', 'medium close-up shot', 'same height as person', 'transitions from frontal view to profile view'], 'style': ['clean', 'minimalist', 'professional']} |

Table 10: Hyperparameters and data sampling ratios for Stage-1 and Stage-2.

| | Stage-1 | Stage-2 | | | |
|---|---|---|---|---|---|
| | | IDs | Depth | Camera | Human Pose |
| Learning rate | $1 \times e^{-5}$ | $2.5 \times e^{-6}$ | $1.5 \times e^{-6}$ | $1.5 \times e^{-6}$ | $1.0 \times e^{-6}$ |
| Batch size per GPU | 6 | 2 | 1 | 2 | 1 |
| Gradient Accumulation Steps | 2 | 2 | 4 | 4 | 4 |
| LR scheduler | Cosine | Constant | Constant | Constant | Constant |
| Weight decay | 0.01 | 0.01 | 0.01 | 0.01 | 0.01 |
| Optimizer | AdamW | AdamW | AdamW | AdamW | AdamW |
| Warm-up steps | 200 | 50 | 50 | 50 | 50 |
| Precision | bfloat16 | bfloat16 | bfloat16 | bfloat16 | bfloat16 |
| Input instruction dropout | 0.4 | 0.4 | 0.4 | 0.4 | 0.4 |
| Max prompt length | 1024 | 1024 | 2048 | 1024 | 2048 |

**Test Dataset.** As shown in Table 12, we primarily evaluate the single- and multiple-condition models on the benchmark introduced in Ju et al. (2025). Additionally, we evaluate our method on VBench (Huang et al., 2024), a widely used text-to-video generation benchmark.

Table 11: Statistics of the training datasets used in Stage 1 and Stage 2.

| Stage | IDs | Depth | Camera | Human-pose | Total |
|---|---|---|---|---|---|
| 1 | 2,124 | 2,066 | 2,185 | 2,077 | 8,452 |
| 2 | 2,057 | 2,041 | 2,177 | 2,034 | 8,309 |

Table 12: Statistics of the constructed test datasets. **#Inst.** denotes the number of instances, and **#Condi.** indicates the number of unique conditions. **Short Cap. #Avg. Len** represents the average caption length of short captions, and **Structured Cap. #Avg. Len.** represents the average caption length of structured captions.

| Type | #Inst. | #Condi. | Short Cap. (#Avg. Len.) | #Structured Cap. (#Avg. Len.) |
|---|---|---|---|---|
| Identities | 200 | 350 | 65.28 | 284.97 |
| Camera | 200 | 200 | 50.25 | 208.01 |
| Depth | 200 | 200 | 54.22 | 225.09 |
| Human Pose | 200 | 200 | 58.38 | 259.03 |
| Camera+Identities | 200 | 622 | 53.41 | 209.17 |
| Camera+Depth | 200 | 400 | 51.43 | 208.81 |
| Identities+Depth | 200 | 555 | 53.14 | 286.83 |
| Camera+Identities+Depth | 200 | 756 | 58.35 | 289.21 |

## D.4 REWARD MODEL ANALYSES.

**Detailed Reward Assignment.** During training, we employ Qwen3-30B-A3B-Instruct-2507 (Yang et al., 2025) as a judging model to evaluate the predicted caption's coverage of these elements. The prompts used for evaluation as follows:

---

**Prompt for Caption Quality Evaluation**

Given two inputs:
- A JSON object that specifies expected entities (objects with attributes), actions, camera descriptions, and style descriptions,
- A reference text,

Your task is to check whether each value in the JSON object is semantically supported by the content in the reference text.
- "Supported" means the caption explicitly or implicitly describes the same fact, even if expressed with different wording (semantic similarity is sufficient).
- "Not supported" means the caption does not mention or imply that fact.
- All attributes of an object must be validated.
- For actions, camera, and style, the same semantic checking applies.

After evaluating all items, compute the overall coverage score = (number of supported values) (total number of values).

Output Requirement:
Return only one number between 0 and 1 representing the overall_coverage score. Do not output explanations, JSON, or any other text.
Format:
Final Score: [overall_coverage score]
=============
Input JSON object: json_data
Input reference text: prediction

---

**The Quality of Reward Model.** To evaluate the stability and reliability of reward estimation, we configured the generation with `max_new_tokens=50`, `temperature=0.3`, and `top_p=1`, and repeated predictions five times for each input. As shown in Fig. 7, the prediction variance has a mean of $3.32 \times 10^{-4}$ and a median of $9 \times 10^{-5}$, both very close to zero, indicating that

repeated evaluations on the same sample exhibit almost no fluctuation. The vast majority of samples show negligible differences across repetitions. Furthermore, the average coefficient of variation is $0.0159$, corresponding to a relative variability of less than $2\%$, which reflects a highly stable scoring behavior. We also observe that the trend line in the Mean Reward vs. Variance plot reveals a negative correlation: higher-reward samples tend to have lower variance, suggesting that the model yields more consistent judgments for high-quality answers. In addition, we compare the predictions of Qwen3-30B Yang et al. (2025) with those of GPT-4o. As illustrated in Fig. 8, Qwen3-30B achieves a mean absolute error of $0.0673$ relative to GPT-4o, indicating a generally close alignment between the two models. Taken together, these results demonstrate that Qwen3-30B serves as a reliable reward assigner, capable of consistently and accurately assessing whether the generated captions are correct.

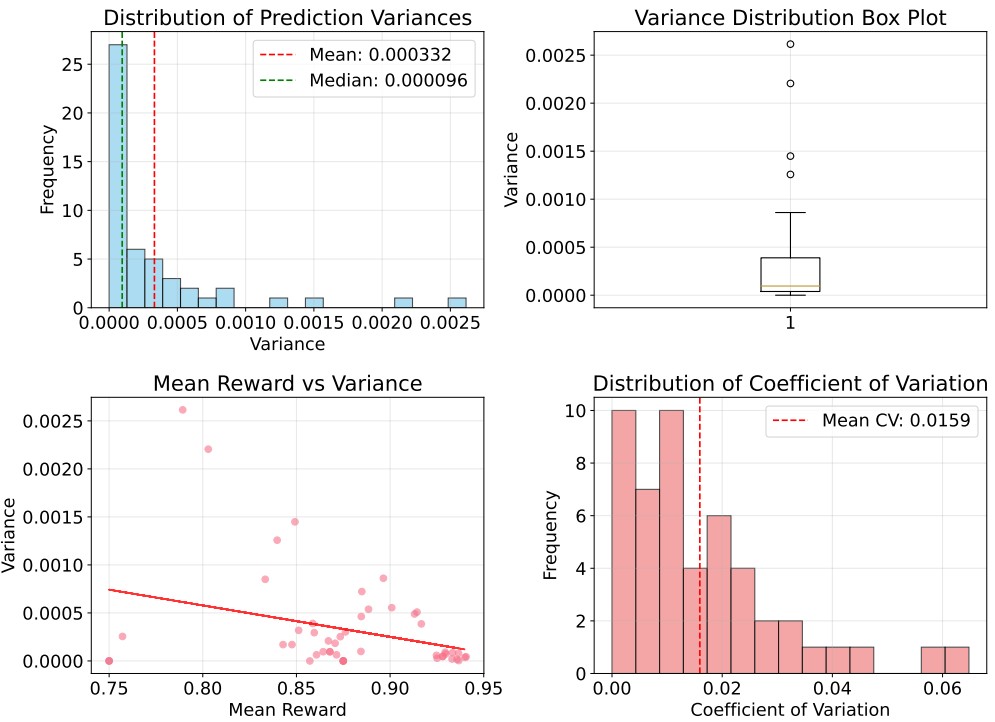

Figure 7: The reward score of consistency analysis of the model Qwen3-30B.

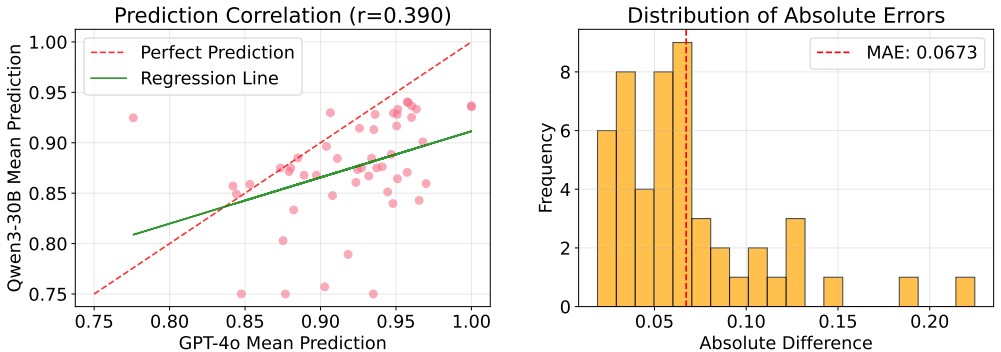

Figure 8: The Qwen3-30B vs. GPT-4o prediction comparison analyses.

Table 13: Comparison between video caption baselines and Instruct-R1 on our proposed benchmark. Baseline models are restricted to video and image inputs; hence, we evaluate them only on the *Depth* and *Multiple Identities* subsets. For fairness, we adopt a one-shot setting, allowing baselines to produce detailed, structured descriptions of target videos. We report the average scores of intention reasoning accuracy (Acc) and quality (Qual.) across all tasks, together with F1 on Dream-1K Wang et al. (2024a).

| Model | Main Object | | Background | | Action | | Style | | Camera | | Other Dataset |
|---|---|---|---|---|---|---|---|---|---|---|---|
| | Acc | Qual. | Acc | Qual. | Acc | Qual. | Acc | Qual. | Acc | Qual. | Dream-1K |
| LLaVA-NeXT-V | 43.57 | 2.13 | 55.41 | 2.16 | 40.87 | 1.72 | 52.64 | 2.43 | 35.97 | 1.90 | 26.10 |
| ShareGPTVideo | 54.90 | 2.68 | 66.77 | 2.65 | 51.92 | 2.05 | 62.48 | 2.94 | 55.29 | 2.68 | 20.40 |
| Qwen2-VL | 51.31 | 2.41 | 63.51 | 2.45 | 50.77 | 1.93 | 59.18 | 2.75 | 51.22 | 2.41 | 29.60 |
| LLaVA-OV | 53.13 | 2.43 | 64.73 | 2.45 | 53.28 | 2.17 | 60.34 | 2.77 | 54.79 | 2.54 | 31.70 |
| Any2Caption | 56.29 | 2.78 | 70.07 | 2.71 | 56.78 | 2.14 | 65.74 | 3.15 | 67.25 | 3.79 | 31.03 |
| Instruct-R1 | **61.38** | **2.89** | **71.07** | **3.04** | **58.78** | **2.34** | **69.76** | **3.83** | **67.25** | **3.79** | **32.45** |

# E  EXTENDED DISCUSSION

## E.1  THE CAPTION CAPABILITY.

We evaluate the interpreter's ability to produce dense, structured captions. Experiments are conducted on our proposed Depth- and multi–IDs–controlled captioning sets, as well as one publicly available benchmark, including Dream-1K Wang et al. (2024a). Results in Table 13 show that our method achieves the best overall intention accuracy and quality score on our condition-controlled datasets. On Dream-1K that emphasizes event-centric descriptions, our model, despite not requiring task-specific fine-tuning, delivers comparable accuracy in capturing movement and event semantics, indicating strong generalization to fine-grained, temporally grounded details.

## E.2  CASE STUDY

First, in Fig. 9 and Fig. 10, we compare videos generated from *original prompts*, *ReaDe-generated prompts*, and *jointly optimized prompts* using feedback from CogVideoX-T2I-2B. ReaDe consistently improves visual fidelity and instruction adherence over the original prompts, and joint optimization with the downstream video generator yields further gains.

Next, Fig. 11 reports side-by-side comparisons across diverse condition types (e.g., identities, pose, depth, camera). In comparison to the original prompts, ReaDe-enhanced prompts yield more coherent compositions, clearer subject–background relations, and improved alignment with the specified controls.

Then, Fig. 12 presents additional qualitative results under a broader set of conditions, demonstrating that ReaDe's prompt refinement generalizes across scenarios and systematically enhances downstream video quality.

Finally, we compare the use of raw prompts with the detailed prompts interpreted by ReaDe, which jointly incorporates both audio and textual inputs. As shown in Fig. 13, the inclusion of audio details leads to higher-fidelity generations and noticeably improved visual quality.

*Original*: A timelapse captures the reaction as concentrated sulfuric acid is poured onto a plastic spoon

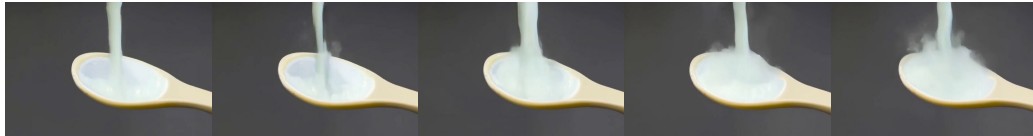

*ReaDe*: A documentary-style timelapse inside a laboratory fume hood. A clean white plastic spoon rests on a glass tray, partially contained within secondary glass shielding, with soft warning labels blurred in the background. The setting is neutral and sterile, with no people visible. A gloved hand briefly pours concentrated sulfuric acid onto the spoon, wetting the bowl and handle. The reaction begins subtly: the smooth surface turns matte, then pits and blisters appear. Over time, the spoon softens, darkens to brown and gray at contact points, and its edges curl and sag. As the degradation accelerates, wispy vapors rise, faint heat shimmer distorts the air, and condensate beads gather on nearby glass before sliding away. The material collapses progressively, leaving a warped, weakened form. The camera remains in a locked medium-to-close shot, beginning just before contact. The sequence transitions into timelapse, with one brief push-in highlighting etched textures, and concludes on a tight close-up of the deformed spoon..

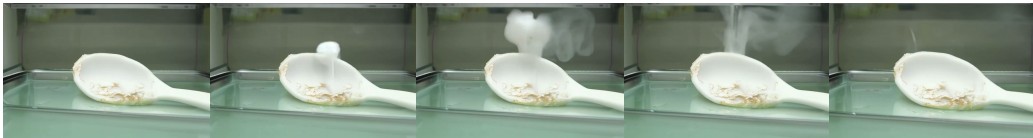

*ReaDe with Video Generator*: Documentary timelapse in a lab fume hood. The sulfuric acid poured onto a clean white plastic spoon on a glass tray, showing progressive degradation. Spoon surface turns matte, then pits, blisters, and softens, browning and curling as edges sag. Neutral lab hood background with glass containment, faint warning labels, no people. Wispy vapors rise, slight heat shimmer, condensate beads slide. Scientific style. The camera locked medium-to-close, starting pre-contact, cutting to timelapse with one push-in on etched texture, ending tight on the warped collapsed spoon.

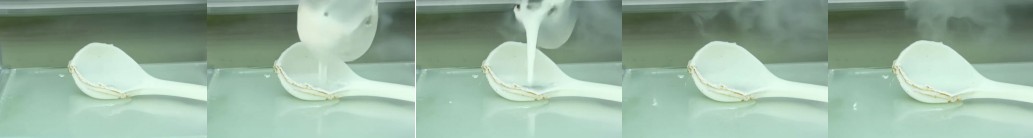

Figure 9: Comparison of videos generated from original prompts, ReaDe-generated prompts, and jointly optimized prompts using feedback from CogVideoX-T2I-2B.

*Original*: Garden, zoom in

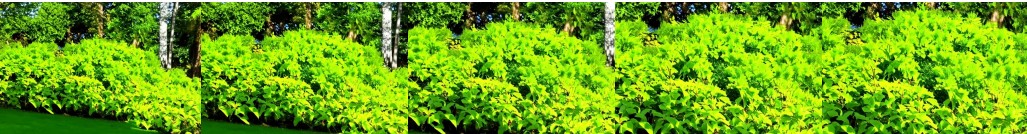

ReaDe: A calm morning, a serene garden is bathed in soft golden light and touched by a thin mist. The atmosphere feels fresh and dewy, with lush greenery and mixed flowers forming a blurred backdrop of warm bokeh. At the center, a single rose bloom glistens with clear dewdrops. A red ladybug crawls slowly along the petal's edge, pausing briefly before continuing its path. A dewdrop gathers and slides down the petal's curve, catching the light as it falls. In the distance, a butterfly drifts across the frame, softly out of focus. The scene moves gently: stems and leaves sway with the breeze, enhancing the sense of quiet vitality. The camera begins with a wide establishing view of the garden, then performs a smooth dolly-in combined with optical zoom, shifting focus from the background to the rose. The sequence concludes in extreme macro, revealing fine petal textures, the glint of the dewdrop, and the ladybug's delicate motion.

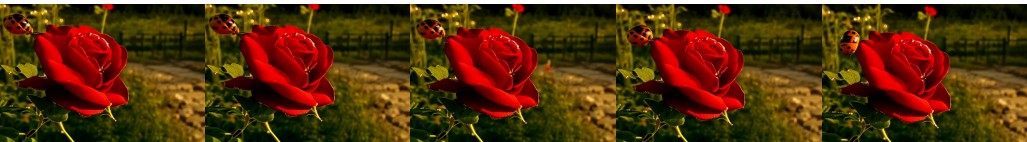

ReaDe with Video Generator: A calm misty morning garden with golden light and dewy freshness. A single red rose bloom holds clear dewdrops, while a ladybug crawls slowly along its petal edge. Blurred greenery, mixed flowers, a faint fence, and a stone path form the background. A dewdrop slides down the petal as the ladybug pauses then moves again; a butterfly drifts softly out of focus. Gentle breeze sways stems and leaves. The camera begins with a wide establishing shot, dolly-in with rack focus to the rose, and ends in extreme macro on the petal and ladybug.

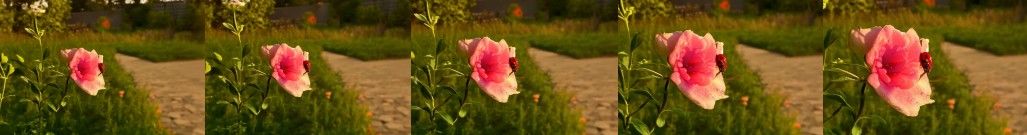

Figure 10: Comparison of videos generated from original prompts, ReaDe-generated prompts, and jointly optimized prompts using feedback from CogVideoX-T2I-2B.

*Original*: Generate a time-lapse that infers morning to afternoon progression.

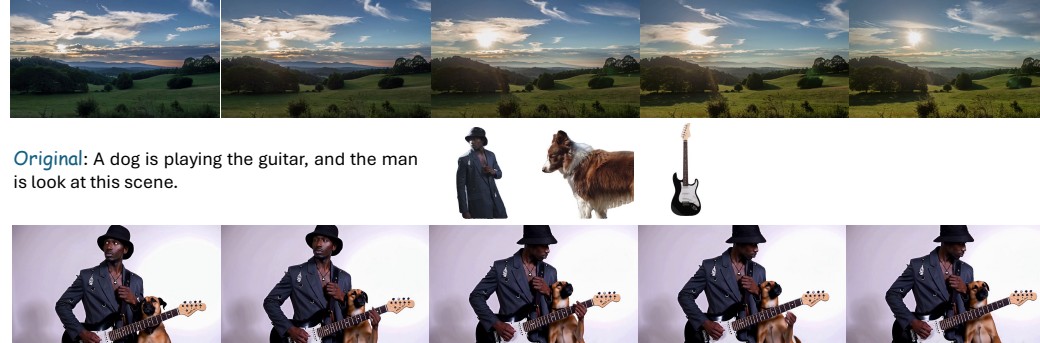

*ReaDe*: A time-lapse video showcases the natural progression from morning to afternoon, highlighting subtle environmental changes over time. The focus remains on a scenic landscape, which may include natural elements such as trees, fields, or a distant skyline. The time-lapse accelerates the movement of shadows, sun path, and any dynamic elements like clouds, illustrating the transition from morning to afternoon. A consistent landscape or cityscape background gradually changes in lighting and shadow as the day progresses from a soft morning glow to brighter afternoon sunlight. The video maintains a smooth, continuous flow with seamless transitions, capturing the essence of a quiet, tranquil day unfolding in fast motion. A fixed camera position steadily captures the wide shot, ensuring all elements in the frame experience a uniform transition in lighting and movement.

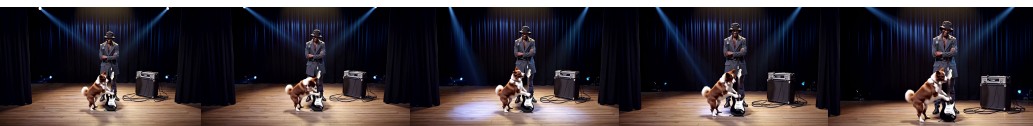

*Original*: A dog is playing the guitar, and the man is look at this scene.

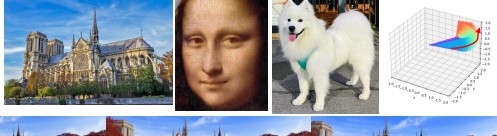

*ReaDe* : The video presents a surreal stage performance under a warm spotlight. At center, a brown-and-white dog stands upright, strumming an electric guitar with rhythmic motions, its head bobbing in time. Beside it, a man watches with folded arms, smiling with quiet amusement. The setting is a simple stage with wooden flooring, a dark curtain backdrop, and faint colored lights. An amplifier and cable sit nearby, adding to the musical scene. Subtle stage beams sweep across as the dog shifts its weight and plays, while the man reacts with small gestures. The camera begins in a medium-wide audience view, slowly dolly-in toward the pair, cuts briefly to a close-up of the dog's paws on the strings, and returns to a wide frame capturing both performer and observer.

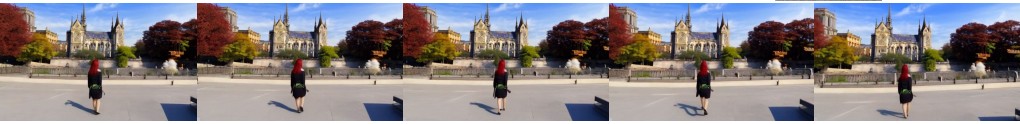

*Original*: A relaxed city stroll: long strides, a splash of red at the head, and a green strap on a small, cloudlike companion; the background is a Gothic front of twin towers and lace-like stonework.

*Original*: A city scene unfolds with a person and a fluffy white dog walking leisurely in front of a grand Gothic cathedral, the person's head adorned with a splash of red, and the dog wearing an eye-catching green harness. The main focus is on a small, cloudlike dog, which is a white Samoyed, with a joyful expression and a green strap harness. Next to it, a human figure is visible, distinguished by a red accessory on their head. Both the person and the dog maintain a relaxed pace with long strides as they stroll comfortably past the historical architecture. The backdrop showcases the imposing facade of a Gothic cathedral, characterized by twin towers and intricate, lace-like stone carvings that add a historical charm to the walking scene. The video maintains a tranquil and timeless ambiance, with a focus on the contrast between the modern walking scene and the historical Gothic architecture, enhanced by natural daylight to emphasize details. The camera keeps a steady, wide-angle shot to encompass both the main subjects and the elaborate architectural details in the background, with occasional gentle pans following the movement while maintaining focus on the entire scene.

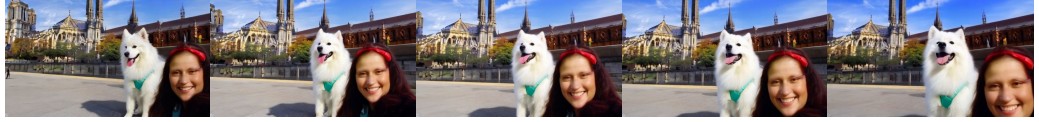

Figure 11: Comparison of videos generated from the original prompts versus ReaDe-generated prompts. Top three rows: Kling1.6; bottom row: FullDiT.

**Prompt**: a video showing the location in the image filled with people who are laughing and chatting 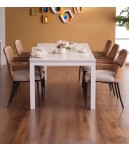

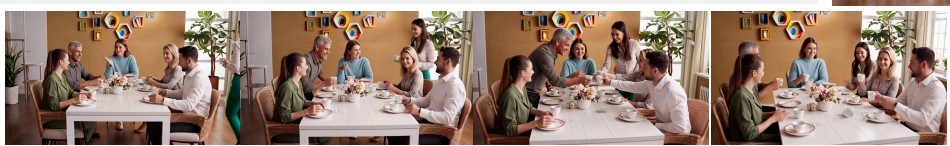

**Prompt** : a video of an animal hunting 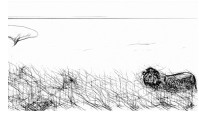

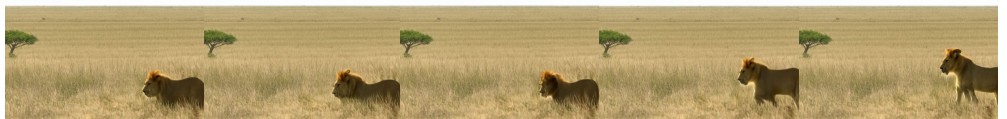

**Prompt** : a smiling young girl with a ponytail walks at a steady pace on a smooth, gently curving empty road. 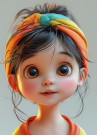 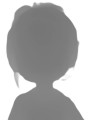 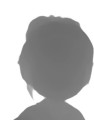 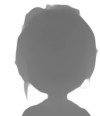

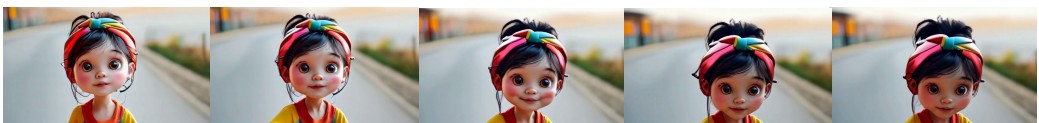

**Prompt** : Peaceful nighttime city street lit by dim streetlights with distant city lights in the background. A sidewalk building displays illuminated "ART CLINIC" signage with its windows and door clearly visible, not obscured. Keep the street empty and quiet, not busy, with minimal traffic 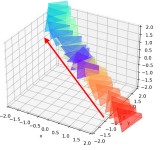

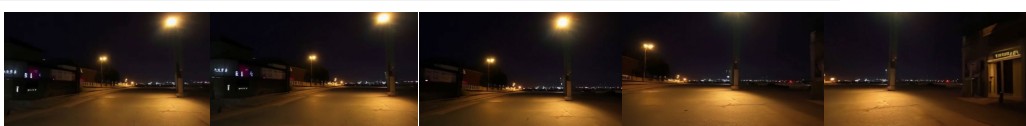

Figure 12: Examples of condition-controlled video generation using ReaDe-extracted detailed prompts. Top row: Kling1.6; second row: SketchVideo; bottom two rows: FullDiT.

Original: A busy street at night.    blaring car horns, the chatter of a noisy crowd

ReaDe: The video depicts a bustling neon-lit city street at night, with rain-slick pavement amplifying the glow of storefronts and billboards. At the center is a crowded intersection, lined with taxis and buses, while the foreground crosswalk is packed with pedestrians carrying umbrellas. A steaming street-food cart anchors the curbside, adding texture to the scene. The background is layered with LED billboards, shop signs, and high-rise windows, their reflections scattered across the wet asphalt. Traffic lights shimmer, distant blue-red strobes flicker, and vapor rises from manholes as paper flyers flutter in the breeze. The soundscape is dense and chaotic: overlapping horns, chatter and laughter, bargaining voices, and the swell of a police siren that grows before fading away. The video emphasizes motion and atmosphere, portraying the restless rhythm of the night. The camera begins in a wide shot from the sidewalk, tracks laterally along storefronts toward the intersection, tilts briefly upward to reveal stacked signs.

Original: People are watching fireworks in the night sky    sound of explosions, the crowd's gasps

ReaDe: The video captures a lively riverside festival night under a clear sky. The central focus is the dazzling fireworks display, with golden chrysanthemums, red and blue peonies, comet tails, and cascading willow trails bursting high above. Each aerial shell blooms into expanding spheres, fades into drifting sparks, and fills the night with echoing thunder. In the background, silhouettes of families gather on blankets, children jump and squeal, and many raise their phones to record the show. Small sparklers flicker near the crowd, while paper lanterns and food stalls add warmth along the riverside. The dark water reflects the brilliant bursts overhead, doubling the spectacle. The atmosphere is filled with layered sounds: booming explosions, crackling trails, overlapping gasps and cheers, children's laughter, and the rhythmic clapping of the audience. People point skyward, shout "wow" and revel in the synchronized volleys of light. Smoke drifts slowly across the scene, caught by the breeze and illuminated by falling sparks. The video adopts a natural, festive style, emphasizing both the grandeur of the fireworks and the joyful responses of the spectators. The camera begins with a wide shot from behind the audience, slowly pushes in while tilting upward toward the sky, pans briefly to follow multiple bursts.

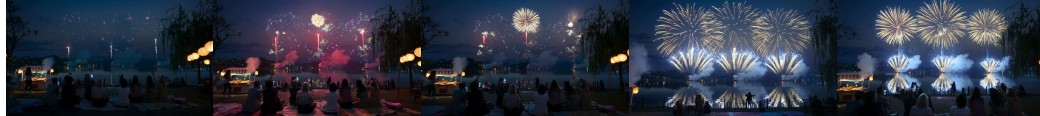

Figure 13: Comparison with/without text-audio conditioned video generation. The results is extracted by Kling1.6.

