# OpenReview forum: "A Reason-then-Describe Instruction Interpreter for Controllable Video Generation"
_ICLR.cc/2026/Conference — ICLR 2026 Conference Withdrawn Submission_

### Official Review · Reviewer_MTFY · 2025-11-01

**Soundness:** 3
**Presentation:** 3
**Contribution:** 3
**Rating:** 4
**Confidence:** 4

**Summary:**

This paper proposes ReaDe, the first universal video instruction interpreter for controllable video generation. It also introduces a multi-dimensional reward assigner to enhance training and control. Experimental results are promising and demonstrate the effectiveness of the proposed approach.

**Strengths:**

The paper proposes ReaDe, the first universal video instruction interpreter for controllable video generation.

The presentation is clear, coherent, and well-structured.

The work introduces new data construction and reward design strategies.

The experimental comparisons are comprehensive and sufficient.

**Weaknesses:**

The paper does not include a reward curve, which is important for illustrating training dynamics and stability.

It remains unclear whether there are any reward conflicts or reward hacking phenomena during optimization.

Fine-tuning an open-sourced model (e.g., Wan) on the refined caption dataset generated by the proposed method could provide a more convincing analysis. Currently, the refined captions are only used during inference, which is not sufficient to fully validate the method’s effectiveness.

**Questions:**

My main concern is that visual control signals such as camera parameters, poses, or reference images cannot always be effectively interpreted as textual prompts. While this approach may work in simpler cases, it is unlikely to generalize well to more complex scenarios. Therefore, the underlying assumption of this line of work—that all forms of visual control can be represented or interpreted textually—does not fully hold.

---

### Official Review · Reviewer_uR5J · 2025-11-01

**Soundness:** 3
**Presentation:** 3
**Contribution:** 2
**Rating:** 4
**Confidence:** 4

**Summary:**

The paper propose a model-agnostic interpreter that converts raw instructions into precise, actionable specifications for downstream video generators.

**Strengths:**

1. The paper is well written.

2. The author uses reinforcement learning to improve the accuracy of the prompts given by the interpreter, and experiments show that better results were achieved.

**Weaknesses:**

1. From an insight perspective, it has been validated by many previous works that accurate prompts lead to better generation results. Therefore, designing a more accurate VLM has not brought additional benefits to this task.

2. The data used is sourced from GPT-4o, so I believe this ability is a distillation of 4o on specific tasks, which isn't particularly interesting.

3. Improving the video captioning ability of VLMs has been the focus of much research before. Moreover, compared to previous methods, using better base models, such as Qwen2.5-omni, will inherently improve model performance. We can also use larger models to achieve improvements.

4.The author is also using CogVideoX-5B, which is somewhat outdated now.

5. I believe the author’s reward should come from feedback on the video output results, rather than the accuracy of the text. Because accurate prompts don't necessarily mean good output video results.

**Questions:**

see weakness

---

### Official Review · Reviewer_TJwC · 2025-11-01

**Soundness:** 3
**Presentation:** 3
**Contribution:** 3
**Rating:** 6
**Confidence:** 3

**Summary:**

This paper introduces ReaDe, a model-agnostic instruction interpreter designed to improve controllability in text-to-video generation. ReaDe follows a reason-then-describe paradigm, using a multimodal LLM to parse user inputs—such as text, images, depth maps, and camera trajectories—into structured, detailed prompts. The method is trained in two stages: supervised fine-tuning with chain-of-thought reasoning, followed by reinforcement learning with multi-dimensional rewards. Experiments show that ReaDe improves instruction faithfulness, caption accuracy, and video quality across single- and multi-condition settings. It also generalizes well to unseen and reasoning-intensive instructions.

**Strengths:**

- The method is well-motivated: it leverages MLLMs to handle multimodal inputs and uses RL to enhance generalization, resulting in more faithful and controllable video generation.
- Comprehensive experiments demonstrate ReaDe’s effectiveness across multiple metrics and condition types, outperforming competitive baselines like Any2Caption.

**Weaknesses:**

- The claim of being *model-agnostic* is insufficiently supported. Quantitative experiments only use CogVideoX-2B; no results are shown for other video generators, limiting the generalizability of this claim.
- The ablation study lacks rigorous comparisons to justify the statement that $R_{\text{user}}$ and $R_{\text{detail}}$ contribute the most. Table 6 does not include controlled experiments that isolate the effect of individual reward components, making the conclusion less convincing.

**Questions:**

- The claim of model-agnosticism raises a conceptual question: if a video model is trained on short captions, could a dense structured prompt from ReaDe hurt its performance? Can the authors provide insight or experiments addressing this?
- The *video-quality feedback* mechanism mentioned in Figure 2 is not described in detail. Could the authors elaborate on how this feedback is implemented and integrated?

---

### Note · Authors · 2025-11-14

I have read and agree with the venue's withdrawal policy on behalf of myself and my co-authors.